# Effect of paternal-maternal parenting styles on college students' internet addiction of different genders: The mediating role of life satisfaction

**Zixin Liu, Hui Cheng, Hexu Guan, Xi Yang, Zi Chen***

School of Psychology, Chengdu Medical College, Sichuan, China

* Zi2116@hotmail.com

**Data Availability Statement:** All relevant data are within the manuscript and its Supporting information files.

## Abstract

This study aimed to understand the current situation of paternal-maternal parenting style, life satisfaction, and internet addiction among college students and explored the influence of paternal-maternal parenting styles and life satisfaction on the internet addiction of male and female college students. A questionnaire survey was administered to 967 college students in China. Life satisfaction partially mediated the effect of the paternal-maternal parenting styles on the internet addiction among college students. However, this mediating role completely varied by gender, and the dimensions of parental styles also had different effects. For male college students, life satisfaction mediated the two dimensions of parenting styles (the father's emotional warmth, the father's overprotection) and internet addiction; the mother's emotional warmth directly related to the internet addiction. Among females, life satisfaction played a partial mediating role between two dimensions of parenting styles (the father's emotional warmth, the mother's interference and protection) and internet addiction. the father's punitiveness and over-involvement were directly related to female students' internet addiction. The study reveals that the mediating effect of life satisfaction on parenting styles and internet addiction among college students is influenced by gender, and the relationship between different parenting styles and internet addiction also varies. These findings indicate that paying attention to the role of the family, especially the parenting style of fathers, is crucial for preventing internet addiction in the future. Prevention and intervention should be treated differently for male and female students.

## Introduction

With the continuous development of the Internet, information technology and the Internet have penetrated into the life and learning of contemporary teenagers. However, improper or excessive use of the internet may lead to addiction [1].

Internet addiction, also known as problematic internet use, pathological internet use or overuse, refers to a tolerant, withdrawal response to the Internet triggered by prolonged inappropriate internet use in which the addict has a persistent desire to access the Internet, thus

**Funding:** The author(s) received no specific funding for this work.

**Competing interests:** The authors have declared that no competing interests exist.

leading to uncontrolled behavior [2]. Once internet addiction occurs [1,3], it might damage an individual's physical, psychological, and social functions [4], leading to depression, anxiety, and insomnia disorders. College students are the susceptible population to internet addiction. A recent meta-analysis study found that about 11.3% of college students in China are addicted to the internet [5]. And another meta-analysis also indicated that the prevalence of internet addiction among Asian college students increased over time [6].

Previous studies on the potential mechanisms of internet addiction have suggested that the mechanisms are influenced by various factors, such as family function [7], peer relationships [8], personal traits [9], etc. Numerous studies have confirmed that the parenting style [10–12] is one of the important factors in predicting college students' internet addiction among family factors. Moreover, previous studies have proven that life satisfaction, as an individual's self-awareness [13], is related to their internet addiction [14,15]. Individuals with lower levels of life satisfaction drive the growth of internet addictive tendencies [15]. Therefore, this study infers that the family parenting style and life satisfaction may be associated with internet addiction of college students. Meanwhile, recent studies have shown that parenting styles are closely related to life satisfaction [16,17]. The way parents educate their children can significantly predict their life satisfaction [18]. Moreover, life satisfaction has been frequently investigated as a mediating factor [19]. Therefore, this study suggests that life satisfaction may mediate between the family parenting style and internet addiction among college students. Furthermore, studies on internet addiction behaviors [20,21], family parenting styles [22] and life satisfaction [23], have found gender difference. Therefore, this study will further explore the differing mechanisms of internet addiction among male and female college students.

According to the cognitive-behavioral model of pathological internet use [24], factors of internet use disorder are divided into distal and proximal ones. Distal factors include life stress and vulnerability; proximal factors include maladaptive cognitive factors. Internet addiction is mainly influenced by distal factors through the mediating effect of proximal factors. As a family environment factor, parenting style is a distal factor in internet addiction, which may influence internet addiction through life satisfaction, a positive cognitive (proximal) factor. This study regards the family parenting style as a distal factor and life satisfaction as a positive cognitive factor to explore their potential mechanism on internet addiction among college students.

## Parenting styles and internet addiction

Parenting styles were divided into two categories, including preference, understanding and authoritative parenting styles, that children are regarded as positive parenting styles [25], while negative parenting styles mainly refer to severe or indulgent parenting styles, such as authoritarian, overprotection, punitive and rejective parenting style [26,27].

Previous studies have shown that different parenting styles can lead to various psychological and behavioral problems such as substance abuse [28] and problematic internet use [11], and the parenting style is an important factor influencing the development of internet addiction [29]. Authoritarian parenting style may lead children to perceive parental supervision as a communication barrier, causing them to seek solace online to avoid stressful family interactions [30]. Ren et.al demonstrated that [31] positive parental behaviors can decrease the risk of children acquiring internet addiction. But recent studies have found that tolerant parenting styles may increase the risk of internet addiction [32,33]. Tolerant and strict fathers may greatly increase the risk of children becoming addicted to the Internet [33]. A child may experience identity confusion and behavioral unpredictability as a result of tolerant parenting, which increases the likelihood of the youngster developing internet addiction [33].

Meanwhile, Liu et al. [34] proposed the "theory of network satisfaction of psychological needs" based on empirical research, stating that when college students' psychological needs are unsatisfied in real life, they seek other ways, like the Internet, to achieve satisfaction. If the Internet can satisfy the needs of such individuals, this may motivate them to use the Internet more often, thus leading to thus leading to problematic internet use. In a lack of warm home environments, adolescents often seek comfort in increased online activities, such as playing online games, which ultimately fuels their growing addiction to the internet [35].

Moreover, several studies [12,35,36] have explored the potential mechanisms of internet addiction from the perspective of specific parenting styles. Lukavská et.al [12] found that the parents' authoritative parenting style decreased problematic internet use. Adolescents whose mother has an authoritarian parenting style and father has a neglectful parenting style had the high prevalence of internet addiction. Parenting styles and internet addiction among adolescents are significantly correlated, according to a meta-analysis [36]. Different parenting styles have different relationships on the young problems (e.g., problematic internet use), exceptionally negative parenting. For example, punitive parenting has a greater impact on adolescent problematic internet use than over-protection and rejection [35]. Some studies are focused on how different parenting styles of fathers and mothers affect adolescents' internet addiction [37–39]. Lansford et.al [37] showed that fathers' psychological control can affect children's developmental problems. Another study [38] revealed that the mother-adolescent relationship is more closely linked to internet addiction than the father-adolescent relationship among adolescents. Fathers are more likely to promote teenage internet addiction with rejective and overprotective parenting styles [39].

Therefore, this study suggests that various dimensions of parenting styles have different effects on the mechanisms of internet addiction among college students.

## The mediating role of life satisfaction

Life satisfaction is a subjective evaluation of an individual's quality of life based on their own standards. Previous studies [40,41] and theory of network satisfaction of psychological needs [34] have suggested that people can use need for life satisfaction to construct cognitive and affective experiences, as life satisfaction is an externalized expression of intrinsic psychological needs that are actually met. Therefore, individuals are more likely to get addicted to the Internet when cognitive and affective experiences become problematic. Similarly, Bozoglan et al. [42] found that life satisfaction is a strong predictor of internet addiction.

Based on the cognitive-behavioral model of pathological internet use, as a positive cognitive factor, life satisfaction is a crucial factor in internet addiction, which may be influenced by distal factor to affect internet addiction [24]. Parents have long been identified as having an important influence on the wellbeing of children and adolescents [43,44]. Similarly, family parenting behaviors may influence a person's level of life satisfaction [45], and both kind and understanding and overprotective parenting may impact college students. Previous study by Trong Dam et al [46] found that adolescents raised in optimistic and positive families had significantly higher life satisfaction indices than those raised in pessimistic and negative families. Adolescents' mental health profits from the positive parenting style, such as parental emotional warmth [47]. In contrast, the mental health of adolescents is negatively associated with parental rejection [48]. Meanwhile, some studies are focused on how different parenting styles of fathers and mothers affect children's life satisfaction [49–51]. Abubakar et al. [51] reported that the authoritative parenting style of fathers rather than mothers positively predicts the life satisfaction of adolescents. However, a study found that [49] in adolescents who perceived a lower paternal authoritarian style, their satisfaction sharply decreased over 2 years in China.

These studies illustrate the differences in influence between mothers and fathers, but there is still controversy.

Moreover, life satisfaction has been frequently investigated as a mediating element [19,52,53]. A study examining the mediating mechanism of life satisfaction between bullying at school and internet addiction [19] found that adolescents who were bullied at school developed negative perceptions and evaluations of themselves and others because of their victimization experiences. Bullied teenagers are more inclined to extend their lives online and possibly develop internet addiction because of a lack of realistic communication in their lives. In summary, this study suggests that life satisfaction may mediate the relationship between parenting styles and internet addiction among college students.

## Gender differences

Previous studies on internet addiction behaviors, family parenting practices, and life satisfaction, have found gender difference [22,23,54,55].

First, multiple studies [20,21,55] have revealed a higher epidemiological prevalence of internet addiction in males than females among adolescents [56]. Discrepancies in specific internet use behaviors associated with internet use have also been found [57]. Male university students spend significantly more time playing online games than female students, and female's problematic internet use behaviors mainly concentrate on chatting and social networking.

Second, research on parenting styles [11] and the Gendered Family Process Model (GFPM) [22] have also suggested that males and females perceive parenting styles differently and that parenting styles have varying effects on the development of boys and girls. Previous research on parent-child relationships and internet addiction has found no gender differences in the parent-child relationships and internet addiction, however their possible influence on internet addiction through non-adaptive cognitions is moderated by gender, with non-adaptive cognitions having a significantly stronger effect on internet addiction for females than for males [58].

Third, research on life satisfaction suggests that there are differences in life satisfaction among adolescents of different genders, but there are still differences at different time periods [59]. Henkens research showed that [60], females report lower levels of life satisfaction and unstable development during adolescence. However, research on life satisfaction suggested that females' life satisfaction decreases from early adolescence and stabilizes or rises from late adolescence to adulthood [23], in accordance with the accelerated maturation hypothesis and stress sensitivity perspective. Although men's life satisfaction also decreases, it tends to start in adulthood and manifest itself slightly later than it does in women. Hence, men frequently have lower levels of life satisfaction as adults than women.

Therefore, this study focuses on gender difference, and aims to explore whether the mechanism of mediation in different gender groups should be different.

## The present study

The relationship of parenting styles and adolescent behaviors has been discussed in the literature, while the mediating mechanisms between parenting styles and young internet addiction are frequently omitted. To prevent or reduce internet addiction, however, and to better understand the relationship between parenting styles and internet addiction, this research is needed. Therefore, this study combines the cognitive behavioral model of pathological internet use to explore the mediating role of life satisfaction as a cognitive factor in the relationship between family parenting styles and internet addiction. The hypotheses are presented below. Fig 1 illustrates the proposed model:

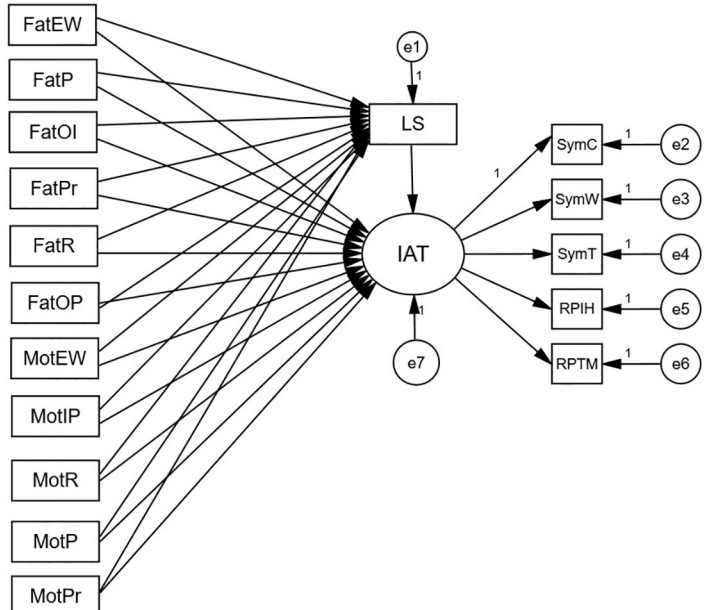

**Fig 1. Theoretical model.**

Hypothesis 1 (H1): The emotional warmth of parenting styles may be negatively correlated with internet addiction, while parenting styles such as overprotection, involvement, punishment, and preference are positively correlated with internet addiction. Life satisfaction has a mediating role in the relationship of parenting styles and internet addiction among Chinese college students.

Hypothesis 2 (H2): The mechanism of mediation in different gender groups should be different.

## Methods

### Participants

This study used a convenient sampling method to randomly select classes from a college in China from September to October 2021, and conducted a questionnaire survey through the psychological organization. Questionnaires totaling 1,048 were distributed, and 620 (64%) female and 338 (35%) male students participated in the survey. The average ages for the whole, male, and female samples were 19.74 ± 1.11, 19.83 ± 1.27, and 19.68 ± 1.01 years, respectively. The data showed a difference in the mean ages ($p < .01$).

This study was reviewed by the Ethics Review Committee of Chengdu Medical College (2021NO.07).

### Measures

Egma Minnen av Bardndosna Uppfostran (EMBU) [27]: The 66-item EMBU was used to assess college students' perceived parenting styles. This scale is divided into two parent sections: The mother's with five factors (emotional warmth, over interference and protection, rejection, punitiveness, and preference) and the father's with six factors (emotional warmth,

punitiveness, over-involvement, preference, rejection, and overprotection). Previous studies [61] classified parenting styles as either positive (emotional warmth) or negative (characterized by punitiveness, involvement, rejection, overprotection, and preference). The scale is graded on a four-point scale ranging from 1 (never) to 4 (always). Each subscale's score is the sum of the corresponding items' scores. Higher subscale scores suggest more frequent usage of a parenting method by the participant's parents. The Cronbach's α for each dimension of this scale are ranging from 0.629 to 0.934. The scale's total Cronbach's α was 0.936. The structural validity of EMBU in the present study was tested by confirmatory factor analysis and results indicated acceptable model fit indices for maternal styles maternal parenting style ($\chi^2/df$ = 3.367, RMSEA = 0.049, GFI = 0.828, SRMR = 0.090, TLI = 0.843), paternal parenting style ($\chi^2/df$ = 3.449, RMSEA = 0.050, GFI = 0.820, SRMR = 0.090, TLI = 0.833).

Satisfaction With Life Scale (SWLS) [62]: This scale comprises five questions on one dimension and based on a seven-point scale ranging from 1 (strong disagreement) to 7 (strong agreement). The final grade is the sum of the scores for the five questions. The Cronbach's α for this scale was 0.896. The structural validity of SWLS in the present study was tested by confirmatory factor analysis and results indicated acceptable model fit: $\chi^2/df$ = 4.439, RMSEA = 0.032, GFI = 0.992, SRMR = 0.014, TLI = 0.989.

Chinese Internet Addiction Scale (CIAS-R) [63]: This scale comprises 26 items, divided into five dimensions (compulsive use of Internet, withdrawal symptoms of internet addiction, tolerance symptoms of internet addiction, interpersonal and health-related problems of internet addiction, and time management problems). It is scored on a four-point scale ranging from 1 (extreme non-conformity) to 4 (extreme conformity). The Cronbach's α for each dimension of this scale are ranging from 0.708 to 0.834. The scale's Cronbach's α was 0.934. The structural validity of CIAS in the present study was tested by confirmatory factor analysis and results indicated acceptable model fit: $\chi^2/df$ = 4.078, RMSEA = 0.056, GFI = 0.907, SRMR = 0.042, TLI = 0.903.

## Procedures

This study contacted professional psychological commissioners through the college psychological organization and collected paper questionnaires offline. Two psychology graduate students and one psychological commissioner from each major jointly distributed and collected questionnaires. The two psychology graduate students read out guidance before distributing the questionnaires, and after distribution, they were collected uniformly. The questionnaire included the instructions, requirements, and precautions. If the number of questions filled in the questionnaire is less than one-third, it will be considered invalid. A total of 1,048 questionnaires were collected: 967 valid (92% recovery rate).

## Data analysis

The results were analyzed using SPSS21.0. An independent samples *t-test* was used to compare gender differences in the variables, and Pearson correlation analysis was applied to examine the relationships among variables. Structural equation modeling (SEM) was developed using AMOS24.0, and the mediating effect of life satisfaction was tested for significance using the bootstrap method. In this study, the reliability and validity tests of each measurement instrument were conducted. When Cronbach's α> 0.6 is considered acceptable and >0.7 is considered greater [64]. Confirmatory factor analysis (CFA) was used to test structural validity. the following model fit indexes were reported: GFI>0.80, TLI>0.80, RMSEA<0.08, $\chi^2/df$<5 and SRMR<0.1 [65–67]. If the above criteria are satisfied, the structural validity fitness is acceptable. A significance level of $p < .05$ was used in this study, and the model fit indicators and

scoring criteria were as follows: $\chi^2/df<2$ (good model); $2<\chi^2/df<5$ (acceptable); GFI, AGFI, NFI, and CFI> .90; RMSEA < .05 (good); and RMSEA < .08 (acceptable) [68]. In this study, Harman's One-Factor test was used to test the Common Method Variance. The explanatory power of the first factor no exceeding the critical value of 50% [69].

## Results

### Common method deviation test

The Harman's One-Factor test was used to conduct a common method deviation test using exploratory factor analysis. The first common factor interpretation percentage was 19.214%, which was less than 50%, suggesting no serious common method bias.

### Basic information about parenting styles, life satisfaction, internet use, and gender differences

Table 1 shows gender differences in all variables. In addition to the dimensions of compulsive use of the internet and tolerance symptoms of internet addiction, there were no gender differences in the total score and other dimensions of internet addiction. Gender differences appeared in all dimensions of fathers' and life satisfaction and mothers' parenting styles and life satisfaction. The life satisfaction score of males was significantly lower than that of females.

### Correlation analysis

Table 2 shows significant associations of parenting style dimensions with life satisfaction level, except for paternal preference. All parenting style dimensions except for parental preference were substantially correlated with the overall score of internet use. All parenting style dimensions except for parental preference were also strongly correlated with each dimension of internet addiction. The mother's preference was substantially correlated with the interpersonal and health-related problems of internet addiction. Internet addiction among students was strongly and negatively correlated with life satisfaction.

### Mediating effect test of the overall model

A final model with a decent fit was obtained after deleting non-statistically significant routes and correcting them with suggestions of MI coefficients. The model's goodness of fit values was as follows: RMSEA = 0.052<0.08, GFI = 0.981>0.9, TLI = 0.977>0.9, NFI = 0.980>0.9, and $\chi^2/df$ = 3.60<5. The final model (Fig 2) demonstrated that life satisfaction has a mediating role in two dimensions of parenting style (the father's overprotection and the father's emotional warmth). Life satisfaction fully mediated the association between the father's overprotection and students' internet addiction while partially mediating the relationship between the father's emotional warmth and students' internet addiction. The students' internet addiction was directly related to maternal over interference and protection.

The mediating effect was further examined using the bias-corrected percentile bootstrap method (5,000 replicate samples), and the results showed (Table 3) that life satisfaction partially mediated the effect of father's emotional warmth and overprotection on college students' internet addiction.

**Table 1. Gender differences in all variables.**

| Variables | All M±SD | Male M±SD | Female M±SD | t |
|---|---|---|---|---|
| | N = 958 | n = 338 | n = 620 | |
| **Father** | | | | |
| FatEW | 51.63±11.30 | 49.11±11.19 | 53.02±11.18 | -5.18*** |
| FatP | 17.66±5.68 | 19.67±6.35 | 16.55±4.97 | 7.82*** |
| FatOI | 18.70±4.12 | 19.83±4.13 | 18.06±3.97 | 6.51*** |
| FatPr | 10.02±3.45 | 9.60±3.27 | 10.28±3.51 | -2.94** |
| FatR | 9.24±2.93 | 10.20±3.10 | 8.71±2.70 | 7.44*** |
| FatOP | 9.84±2.47 | 10.09±2.41 | 9.67±2.49 | 2.54* |
| **Mother** | | | | |
| MotEW | 52.31±11.38 | 49.63±11.31 | 53.77±11.22 | -5.44*** |
| MotIP | 31.72±6.68 | 33.09±6.45 | 30.91±6.67 | 4.90*** |
| MotR | 12.57±3.97 | 13.57±4.07 | 11.99±3.78 | 5.90*** |
| MotP | 12.97±4.36 | 14.26±4.70 | 12.25±4.01 | 6.68*** |
| MotPr | 11.04±3.00 | 10.66±2.94 | 11.29±3.00 | -3.13** |
| LS | 20.25±6.31 | 19.68±6.42 | 20.52±6.21 | -1.98* |
| IA | 58.57±11.60 | 58.01±12.45 | 58.93±11.08 | -1.15 |
| SymC | 10.91±2.55 | 10.63±2.64 | 11.07±2.49 | -2.58* |
| SymW | 11.67±2.52 | 11.47±2.68 | 11.80±2.43 | -1.95 |
| SymT | 9.63±2.08 | 9.39±2.20 | 9.78±1.99 | -2.73** |
| RPIH | 15.32±3.58 | 15.33±3.81 | 15.33±3.45 | -0.03 |
| RPTM | 11.03±2.77 | 11.19±2.89 | 10.94±2.69 | 1.33 |

Note:

* denotes $p < .05$,

** denotes $p < .01$,

*** denotes $p < .001$.

FatEW = Father's emotional warmth, FatP = Father's punitiveness, FatOI = Father's over-involvement, FatPr = Father's preference, FatR = Father's rejection, FatOP = Father's overprotection, MotEW = Mother's emotional warmth, MotP = Mother's punitiveness, MotIP = Mother's interference and protection, MotR = Mother's rejection, MotPr = Mother's preference, LS = Life satisfaction, IA = Internet addiction, SymC = Compulsive internet use; SymW = Withdrawal symptoms of internet addiction; SymT = Tolerance symptoms of internet addiction; RPIH = Interpersonal and health-related problems of internet addiction; RPTM = Time management problems.

The same as below.

## Life satisfaction as a mediator of parenting style and student internet addiction: A male-female comparison

According to previous studies, males and females differ significantly in terms of family parenting style, life satisfaction, and internet addiction behavior. This study compared and analyzed the male and female subgroups based on the overall theoretical composition to determine whether differences exist in the mediation mechanisms among college students of different genders. After deleting non-statistically significant paths and correcting by the MI coefficients, two models with better fit were identified. All paths in the model were statistically significant ($p < .05$; see Table 4).

Figs 3 and 4 show the final models among male and female college students, respectively. These figures reveal that the parental dimensions' impact and the mediating role of life satisfaction completely differed by gender. Among males, life satisfaction played a full mediating role on the effect of the two dimensions of father (emotional warmth and overprotection) on

**Table 2. Bivariate correlation analysis of the study variables.**

| Variables | LS | IA | Sym-C | Sym-W | Sym-T | RP-IH | RP-TM |
|---|---|---|---|---|---|---|---|
| FatEW | 0.31** | -0.26** | -0.27** | -0.16** | -0.17** | -0.29** | -0.22** |
| FatP | -0.14** | 0.23** | 0.20** | 0.14** | 0.15** | 0.26** | 0.21** |
| FatOI | -0.14** | 0.16** | 0.17** | 0.09** | 0.09** | 0.18** | 0.13** |
| FatPr | 0.05 | -0.04 | -0.04 | 0.00 | -0.03 | -0.06 | -0.02 |
| FatR | -0.19** | 0.22** | 0.19** | 0.13** | 0.15** | 0.24** | 0.19** |
| FatOP | -0.14** | 0.15** | 0.14** | 0.10** | 0.14** | 0.15** | 0.12** |
| MotEW | 0.28** | -0.23** | -0.24** | -0.13** | -0.14** | -0.26** | -0.18** |
| MotIP | -0.19** | 0.22** | 0.20** | 0.13** | 0.17** | 0.23** | 0.19** |
| MotR | -0.22** | 0.24** | 0.23** | 0.12** | 0.19** | 0.27** | 0.20** |
| MotP | -0.17** | 0.23** | 0.22** | 0.14** | 0.16** | 0.25** | 0.18** |
| MotPr | 0.10** | -0.04 | -0.05 | 0.01 | -0.03 | -0.06* | -0.02 |
| LS | 1.00 | -0.27** | -0.24** | -0.17** | -0.23** | -0.27** | -0.23** |

Note:

* denotes $p < 0.05$;

** denotes $p < 0.01$.

internet addiction; and one dimension of mother (emotional warmth) was directly related to males' internet addiction (Fig 3). Among females, life satisfaction played a partial mediating role on the effect of the two dimensions of parents (the father's emotional warmth and the mother's over interference and protection) on internet addiction; two dimensions of father (punitiveness and over-involvement) were directly related to females' internet addiction (Fig 4).

Mediating effects across gender were further examined using a bias-corrected percentile bootstrap method (5,000 replicate samples). The results showed (Table 3) life satisfaction partially mediated the relationship between the father's emotional warmth and internet addiction, and life satisfaction fully mediated the impact of the father's overprotection on internet addiction among males. Life satisfaction played a mediating role in the relationship between the father's emotional warmth and the mother's interference and protection and internet addiction in females, with both direct and indirect effects being significant. In short, life satisfaction played a partial mediating role.

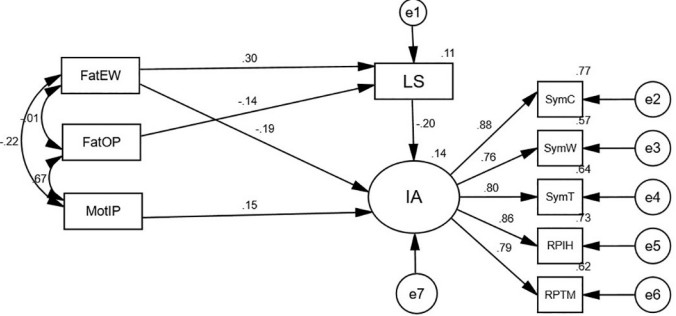

**Fig 2. Impact of parenting style on internet addiction: The mediation model of life satisfaction.**

**Table 3. Mediation effect test.**

| | | Effect value | S.E | 95% C.I. |
|---|---|---|---|---|
| Total | FatEw→ IA | -0.21 | 0.03 | [-0.27, -0.14] |
| | FatEW→ LS→ IA | -0.07 | 0.01 | [-0.09, -0.04] |
| | FatOP→ IA | 0.55 | 0.15 | [0.26, 0.83] |
| | FatOP → LS→ IA | 0.17 | 0.05 | [0.08, 0.26] |
| Male | FatEw→ IA | -0.23 | 0.06 | [-0.34, -0.11] |
| | FatEW→ LS→ IA | -0.08 | 0.03 | [-0.13, -0.03] |
| | FatOP→ IA | 0.40 | 0.27 | [-0.13, 0.93] |
| | FatOP → LS→ IA | 0.25 | 0.11 | [0.07, 0.48] |
| Female | FatEw→ IA | -0.22 | 0.04 | [-0.30, -0.14] |
| | FatEW→ LS→ IA | -0.05 | 0.02 | [-0.09, -0.02] |
| | MotIP→ IA | 0.33 | 0.07 | [0.20, 0.46] |
| | MotIP→ LS→IA | 0.07 | 0.02 | [0.03, 0.11] |

**Table 4. Model fit comparison.**

| Model | $\chi^2/df$ | GFI | TLI | NFI | CFI | RMSEA |
|---|---|---|---|---|---|---|
| Males | 2.145 | 0.965 | 9.979 | 0.971 | 0.984 | 0.058 |
| Females | 2.830 | 0.976 | 9.972 | 0.975 | 0.983 | 0.054 |
| Indices | <5 | >0.90 | >0.90 | >0.90 | >0.90 | <0.08 |
| | acceptable | good | good | good | good | acceptable |

## Discussion

### The current situation between college students' paternal-maternal parenting style, life satisfaction, and internet addiction

The results revealed that female college students scored significantly higher than male college students on the family parenting styles' positive dimensions, whereas males scored significantly higher on the negative dimensions. These findings suggest that males (females) are more likely to grow up in harsh and negative (warm and positive) families. The traditional Chinese concept of "Sons be raised in frugality and daughters in abundance" may help explain

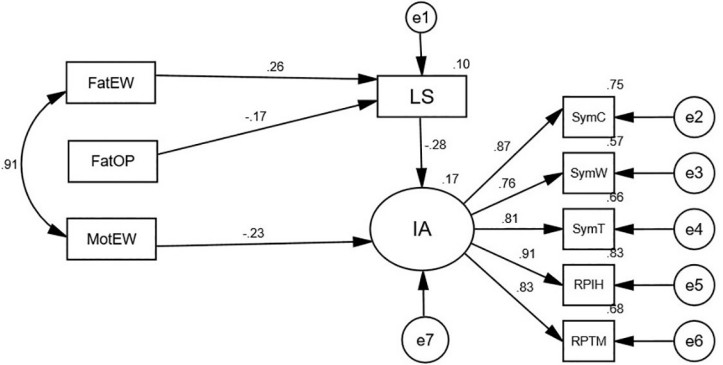

**Fig 3. Final model of male.**

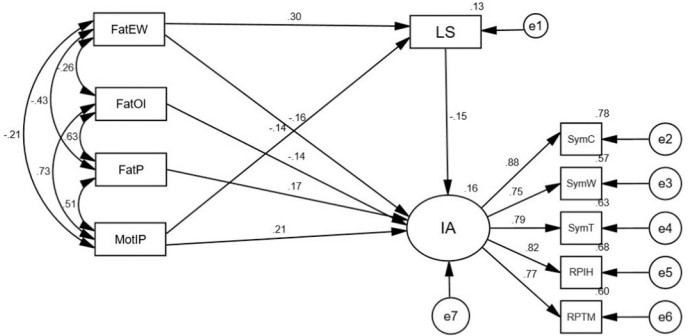

**Fig 4. Final model of female.**

this—in contrast to females, who are more likely to feel loved by their parents, have their material and emotional needs satisfied, and receive a positive and warm parenting style that promotes the development of their personality and personal skills, males are more likely to grow up with a free-range parenting style. This study supports earlier research [23,59,70] about gender differences in life satisfaction by showing that females report higher levels of life satisfaction than males. This might be a result of the psychological maturity difference between males and females, with females being more psychologically developed, typically more psychologically resilient than males, and possessing more coping mechanisms. In contrast to the majority of earlier studies, no gender difference was found in the respondents' levels of internet addiction [20,55,56,71]. This may be caused by the different problematic internet use behaviors among males and females. Males play online games more than females, and females engage in online social interactions and shopping more than males [72]. However, this study also found that females scored significantly higher than males in the compulsive internet use and tolerance dimensions. This suggests an increase in internet addiction behaviors among females. The Internet provides faster and more convenient communication, shopping, and entertainment, and Chinese women spend a lot of time interacting with friends and family on social applications and shop online using applications like Taobao. These indicate a gradual increase in various online behaviors among females; hence, female college students may also become addicted to the Internet through these online behaviors.

## Mediating role model

This study also examined the mediating mechanism of life satisfaction on the relationship of parenting style and internet addiction among college students. The result demonstrated that life satisfaction has a mediating role in two dimensions of parenting style (the father's overprotection and the father's emotional warmth). Then the H1 was partly supported.

First, the results showed that the father's emotional warmth could positively associate with internet addiction behaviors directly and indirectly by the mediating role of life satisfaction. However, a previous study concluded that a father's permissive parenting may lead to identity confusion, likely causing children to become internet addicts [33]. Conversely, the present study revealed that college students who experience their fathers' tolerance and understanding were less likely to become addicted. The father is essential to the growth of adolescents, and their moral influence has a lasting effect far into adulthood. In addition, college students who perceive their parents as kind and understanding are less likely to become internet addiction, but too much tolerance or understanding might be construed as pampering. Future research

on the relationship between warm parenting styles and children's internet addiction should focus on the degree of parental tolerance for adolescents. Previous studies have found that maternal emotional warmth and understanding are protective factors for internet addictive behaviors, and parental rejection can exacerbate adolescent internet addictive behaviors [73,74]. However, maternal warmth perceived by college students and parental rejection were not found to exacerbate the participants' internet addiction.

Second, maternal interference and protection was directly related to a person's internet addiction. Li et.al demonstrated that parental over protection may influence problematic behaviors, such as internet addiction, in adolescents [74]. In the present study, only maternal interference and protection directly influenced internet addiction; college students perceived that their fathers' overprotection do not directly exacerbate their internet addiction but rather influenced internet addiction through the mediating effect of life satisfaction. Overly protective mothers may impact internet addiction because mothers frequently contribute to the development of a stable home environment [75]. Third, overprotective fathers indirectly associated with college students' internet addiction. Harsh parenting represents excessive parental interference and control at home. Adolescents who experience harsh parenting may interact negatively with their parents and others, making them more likely to form negative opinions of others and of reality, which may lead to internet addiction [76].

### Gender differences in the mediating role model

A key finding of this study is the significant gender differences in the mediation mechanisms of life satisfaction on the effect of parenting style on internet addiction and H2 was supported.

Among male students, life satisfaction partially mediated the relationship between fathers' emotional warmth, maternal overprotection and internet addiction; the father's punitiveness directly related to the participants' internet addiction. Previous studies have revealed that parental emotional warmth and the father's overprotection are predictors of male students' internet addiction [11,77]. The present study found that the mothers' tolerant style protected males from becoming internet addicts, whereas the fathers' parenting style indirectly influenced men's internet addiction behavior through the mediating role of life satisfaction. According to Bronte-Tinkew et al. [78], strict fathers contribute to adolescents' risk for criminal behavior and drug use. A strict parenting style may cause children to perceive parental supervision as a barrier to communication, and youngsters who spend more time online to avoid difficult family interactions are more likely to display indicators of internet addiction [77]. However, social class psychology research has discovered that life satisfaction is positively predicted by family socioeconomic status, and better family socioeconomic status typically begets higher life satisfaction [79]. Conversely, in traditional families, the father is the main breadwinner, and students' living expenses mainly come from their families. Thus, an individual's level of life satisfaction is more closely associated with the father. An increase or decrease in a student's monthly living costs decided by the father may impact the life satisfaction of the student, who may turn online as compensation. The mechanism by which a mother's care impacts men's internet addiction may be connected to attachment theory [80], which holds that if a caregiver is caring and helpful during one's childhood, the child will grow up with a positive perception and consider both themselves and their caregivers as trustworthy and reliable. These motivating beliefs progressively become generalized and assimilated, thus changing the child's behavior over time.

For female participants, the present study's findings mostly concur with the previous studies [11,75]: The father's emotional warmth and the mother's interference and protection not only directly exacerbated female students' internet addiction but also indirectly associated

with it through life satisfaction. Mothers frequently foster a positive home environment for adolescents. However, too many limitations or a lack of support from the mother may cause a youngster to become an internet addict [41]. Several previous studies also concluded that harsh parental punishment could predict internet addiction [35,41,81]. Second, the present study found that a father's harsh punishment may detrimentally related to female students' internet addiction. Less communication and connection between parents and children, and increased family conflict have been found in students who perceive harsh parental punishment [82,83]. Students are more likely to use the Internet because they find it difficult to receive the desired emotional response, attention, and care from their parents [84]. The negative parenting style of fathers directly associated with the risk of internet addiction among female college students, which is consistent with Zhang et al. [35]. If adolescents experience punishment from their parents, they are more likely to become Internet addict. The negative correlation between father's over-involvement and female internet addiction may be related to cultural background. China is a typical collectivist cultural country, and in this context, psychological control is often seen as a manifestation of parental care. Adolescents have a relatively high level of acceptance of control intervention behavior. The results indicated that compared with male students, female students' internet addiction was influenced by a variety of circumstances, including their parents' parenting styles both directly and indirectly through the psychological aspects.

The current study found that fathers, more than mothers, significantly influenced their female or male offspring's behavior; only the mothers' emotional warmth and over interference and protection associated with their internet addiction behavior. Previous studies found that the father's parenting style is more crucial to children's development than the mother's [78,85]. Bronte-Tinkew et al. [78] indicated that the father-child relationship has a considerable impact on risky teenage behavior as children outgrow, whereas fathers play a more essential role in adolescent development [86]. The results indicate the need for further research on the underlying mechanisms through which fathers impact adolescents' addictive behavior, and suggest that fathers should be cognizant of how their parenting style affects their adolescents' problematic behavior. This study might be helpful for parents to rethink about their parenting styles, such as improving or adjusting their parenting behavior to reduce the risk of adolescent internet addiction. More specifically, parents should care about and support adolescents, develop positive interactions with them, praise them. Meanwhile, the influence of parenting styles on college students' internet addiction is also influenced by the self-awareness factor of life satisfaction. This suggests that in the future, prevention and intervention for college students' internet addiction should not only involve parental intervention, but also help college students improve their own life satisfaction, for example, helping to improve emotional regulation, relax, and increase effective interpersonal communication. Moreover, the prevention and intervention of internet addiction in both male and female college students deserve different treatments: When focusing on the work of both males and females, attention should be paid to the impact of different parenting dimensions on individuals' internet addiction. In conclusion, this study suggests that in the future, more attention should be paid to the role of family factors in solving the internet addiction among college students. Starting from the cooperation between families and schools, the prevalence of internet addiction among college students may be reduced.

## Limitations

Despite the study's contribution to the literature, it is subject to some limitations. First, the sample was concentrated in one college in China; hence, the possibility of geographical

differences in the sample cannot be ruled out. Future research will expand the sample size and collect samples from multiple universities and wider regions. Second, this study employed a cross-sectional design that made it difficult to infer causality. Future researchers may use longitudinal designs and panel study to explore the underlying mechanisms. Finally, the internet addiction measurement tool used here does not differentiate between various forms of internet addiction (Internet shopping, social media, Internet game), and given the current state of internet addiction, future research may consider specific differentiation to explore the different internet addiction behaviors that are likely to be affected.

## Conclusion

This study mainly explored the potential mechanisms of internet addiction, the impact of family parenting styles on college students' internet addiction, the mediating role of life satisfaction, and the role of gender in it. Parenting styles, particularly that of the father, have a major effect on students' internet addiction, which is partially mediated by life satisfaction. For students of different genders, life satisfaction has different mediating effects, and the influence of parenting styles also present different forms. These findings indicate that paying attention to the role of the family, especially the parenting style of fathers, is crucial for preventing internet addiction in the future. This study may help parents rethink their parenting styles, such as improving or adjusting their parenting behavior based on gender, to reduce the risk of adolescent internet addiction.

## Supporting information

**S1 File.**
(DOCX)

**S1 Raw data.**
(SAV)

## Acknowledgments

The authors thank Chengdu Medical College for providing the participants. Also, the authors thank Editage (www.editage.cn) for English language editing.

## Author Contributions

**Data curation:** Zixin Liu, Hexu Guan.

**Investigation:** Hui Cheng, Xi Yang.

**Writing – original draft:** Zixin Liu.

**Writing – review & editing:** Zixin Liu, Zi Chen.

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
