## [Decision Letter · Decision Letter 0]

8 Feb 2024

PONE-D-23-36156Effect of paternal-maternal parenting styles on college students’ internet addiction of different genders: The mediating role of life satisfactionPLOS ONE

Dear Dr. Chen,

Thank you for submitting your manuscript to PLOS ONE. After careful consideration, we feel that it has merit but does not fully meet PLOS ONE’s publication criteria as it currently stands. Therefore, we invite you to submit a revised version of the manuscript that addresses the points raised during the review process.

**Be sure to:**

**-  reply to each comment from the reviewers. **

Please submit your revised manuscript by Mar 24 2024 11:59PM. If you will need more time than this to complete your revisions, please reply to this message or contact the journal office at plosone@plos.org. Please include the following items when submitting your revised manuscript:A rebuttal letter that responds to each point raised by the academic editor and reviewer(s). You should upload this letter as a separate file labeled 'Response to Reviewers'.A marked-up copy of your manuscript that highlights changes made to the original version. You should upload this as a separate file labeled 'Revised Manuscript with Track Changes'.An unmarked version of your revised paper without tracked changes. You should upload this as a separate file labeled 'Manuscript'.

We look forward to receiving your revised manuscript.

Kind regards,

Fadwa Alhalaiqa

Academic Editor

PLOS ONE

Journal Requirements:

2. In the online submission form, you indicated that The datasets generated and analyzed during the current study are not publicly available but are available from the corresponding author.

Reviewers' comments:

Reviewer's Responses to Questions

**Comments to the Author**

1. Is the manuscript technically sound, and do the data support the conclusions?

Reviewer #1: Partly

Reviewer #2: Yes

2. Has the statistical analysis been performed appropriately and rigorously? 

Reviewer #1: Yes

Reviewer #2: Yes

3. Have the authors made all data underlying the findings in their manuscript fully available?

Reviewer #1: No

Reviewer #2: No

4. Is the manuscript presented in an intelligible fashion and written in standard English?

Reviewer #1: No

Reviewer #2: Yes

5. Review Comments to the Author

Reviewer #1: Review Comments

Thank you very much for the opportunity to participate in reviewing your manuscript, which is an interesting topic. But I still have some concerns that need to be addressed, otherwise the manuscript will not be published in PLOS ONE:

Abstract:

1.Please unify the key terms. Is it Internet use, Internet use behavior or Internet addiction? These are several different concepts and terms. Please check the abstract and the full manuscript carefully. Please use one special term uniformly. This is a very serious matter.

2.Lines 21 – 22 of the abstract, is there a grammatical mistake in this sentence? I don't understand.

Introduction：

3.The quotation of the whole introduction part is not in line with the academic norms, and the non-personal views should be supported by the literature.

4.Lines 41-42, as far as I know, there is not a few literature related to parenting style, life satisfaction and Internet addiction. Please cite the appropriate literature of recent years to support this statement.

5.The logic of the Introduction needs to be adjusted and supplemented, and several subheadings need to be clarified to clearly state the relationships between the various variables, the theoretical foundations, and the hypotheses of the study.

6.Lines 97 and 99, “Previous studies........ , multiple studies......”, but to cite only one study as support is very loose. This kind of situation occurs in many places in the manuscript, and this problem is very bad and needs to be paid attention to by the author.

7.Please clarify the research questions and the study hypotheses.

8.If the hypothesis model is to explore the dimensions, the literature review on the relationship between the dimensions and the main variables needs to be supplemented in the Introduction.

Methods

9.The data for this study were collected from September to October 2021, and it is currently 2024, so the timeliness of the data is worth considering. In addition, what is the serial number of ethical approval?

10.The data is collected through the psychological system. Is there a valid link to this system?

11.Please provide sample items for each scale.

12.Please specify the process of questionnaire collection and how to identify invalid questionnaires?

13.Authors only provide reliability, whether the validity meets the criteria, please add.

Results

14.Please provide valid references for the standard values of all indicators in the Results section, such as Common method deviation test (180 lines), etc.

Others:

15.Please add practical suggestions.

16.Please specify exactly how future research should proceed in the Limitations.

17.The findings, contributions and implications of this study need to be summarized in the concluding section.

In short, there are many errors in the translation of the manuscript that need to be improved, no DOI number in many references (for example, [14], [21]), confusion of proper nouns, the manuscript needs to be carefully revised, and the current quality does not meet the academic level of PLOS ONE journals.

Reviewer #2: No strong arguments for mediation are presented. Moreover, hypotheses are missing. Each parental style may be defined and linked with the other main study variables. In the introduction, there are several information about authoritarian, tolerant, positive parental behaviors. However, the instrument measure emotional warmth, punitiveness, over-involvement, preference, rejection, and overprotection, but not reference to these styles and the relation with life satisfaction and internet addiction is made in the introduction. Moreover, the differential role of mother and father in predicting life satisfaction and internet addiction should also be discussed in the Introduction.

For what reasons some questionnaires are considered invalid?

For the instruments, Alpha Cronbach need to be reported for each scale. Moreover, the results of factorial analysis for the multidimensional scales should be presented in the Method section.

More clarity is needed in the presentation of the Results. This section can be organized according to the study objectives and hypotheses.

6. PLOS authors have the option to publish the peer review history of their article (what does this mean?). If published, this will include your full peer review and any attached files.

Reviewer #1: **Yes: **Jun Li

Reviewer #2: No

---

## [Author Response · Author response to Decision Letter 0]

12 Mar 2024

To Editor,

We sincerely thank the editor and all reviewers for valuable feedback that we have used to improve the quality of our manuscript. According to your nice suggestions, we have made extensive corrections to our previous manuscript, the detailed corrections are listed below. The reviewer comments are laid out below in bold font and specific concerns have been numbered. And changes/additions to the manuscript are identified using the “Track Changes” option in Microsoft Word.

#Reviewer 1:

Comments 1: Please unify the key terms. Is it Internet use, Internet use behavior or Internet addiction? These are several different concepts and terms. Please check the abstract and the full manuscript carefully. Please use one special term uniformly. This is a very serious matter.

Response to comments 1:

Thank you for pointing out the omissions in the manuscript. We have carefully revised and unify the concept. In this study, the main research variable was internet addition, therefore, we have carefully revised the manuscript regarding word usage errors.

Revised content: 

See lines: 14, 18, 23, 27, 29, 66, 123, 166, 198, 201, 207, 211, 214, 220, 221, 223, 465, 496, 501-503, 510, 605, 607. (revised manuscript with track changes) 

See lines:14, 18, 23, 25, 27, 53, 100, 129, 150, 153, 158, 161, 164, 169, 171, 173, 378, 408, 412-414, 421, 506, 507. (manuscript)

Comments 2: Lines 21 – 22 of the abstract, is there a grammatical mistake in this sentence? I don't understand.

Response to comments 2:

We would like to thank the reviewer for pointing out this issue. The error in lines 21-22 has been modified. We have changed this sentence to: For male college students, life satisfaction mediated the two dimensions of parenting styles (the father’s emotional warmth, the father’s overprotection) and internet addiction.

Revised content: 

See lines:20-23. (revised manuscript with track changes)

See lines:20-22. (manuscript)

Comments 3: The quotation of the whole introduction part is not in line with the academic norms, and the non-personal views should be supported by the literature. 

Response to comments 3:

We sincerely appreciate the valuable comments. We have checked the literature carefully. Moreover, we have added more references into Introduction to support this idea.

Comments 4: Lines 41-42, as far as I know, there is not a few literature related to parenting style, life satisfaction and Internet addiction. Please cite the appropriate literature of recent years to support this statement. 

Response to comments 4:

As suggested by the reviewer, we have adjusted the Introduction by deleting lines 41-42 and moving them to lines 60-64, 64-66 (revised manuscript with track changes; see lines 47-50, 51-54 in manuscript). Moreover, we have added some new references about to support this statement.

Comments 5: The logic of the Introduction needs to be adjusted and supplemented, and several subheadings need to be clarified to clearly state the relationships between the various variables, the theoretical foundations, and the hypotheses of the study.

Response to comments 5:

Thank you for your suggestions on improving the manuscript. We have reorganized the logic based on the original manuscript, carefully adjusted and modified the structure and logic of the Introduction of the manuscript. In the Introduction, we have introduced the research objective, and then elaborate on the main theoretical model used in this study. Then, we have explored the current situation of parenting styles and internet addition, the mediating role of life satisfaction, and gender differences. Finally, we have integrated the previous discussion and propose the hypothesis of this study in The present study part. In The present study, we proposed two hypotheses for this study: Life satisfaction has a mediating role in the influence relationship of parenting styles on internet addiction among Chinese college students (H1). The mechanism of mediation in different gender groups should be different. (H2).

Comments 6: Lines 97 and 99, “Previous studies........ , multiple studies......”, but to cite only one study as support is very loose. This kind of situation occurs in many places in the manuscript, and this problem is very bad and needs to be paid attention to by the author.

Response to comments 6:

We are sorry for our careless mistakes. We have carefully revised the relevant parts according to the manuscript, some adding more references to support, and some modifying the words.

Add references: 

See lines: 63, 70, 75, 114, 131, 145, 160, 197, 209, 210, 464, 547, 569. (revised manuscript with track changes)

See lines: 50, 57, 61, 91, 103, 113, 123, 149, 159, 160, 371, 377, 458, 474. (manuscript)

Revised: 

Lines 109, 497. (revised manuscript with track changes)

Lines 89, 410. (manuscript)

Comments 7: Please clarify the research questions and the study hypotheses.

Response to comments 7:

We have carefully revised this section and added a new part, The present study, to clarify the hypotheses of this study. In The present study, we proposed two hypotheses for this study: Life satisfaction has a mediating role in the influence relationship of parenting styles on internet addiction among Chinese college students (H1). The mechanism of mediation in different gender groups should be different. (H2).

Revised content:

See lines:243-258. (revised manuscript with track changes)

See lines:187-198. (manuscript)

Comments 8: If the hypothesis model is to explore the dimensions, the literature review on the relationship between the dimensions and the main variables needs to be supplemented in the Introduction.

Response to comments 8:

Thank you for pointing out the shortcomings of our literature review. We have carefully revised and added the literature review on dimensions and main research variables. The literature review on the relationship between dimensions of family parenting styles and internet addiction is in Lines 131-144 (revised manuscript with track changes); The literature review on the dimensions and life satisfaction is in Lines 174-176, 181-183 (revised manuscript with track changes). And we added more references on the relationship between the dimensions and the main variables.

Revised content: 

See lines:103-112, 134-136, 138-141. (manuscript)

Comments 9: The data for this study were collected from September to October 2021, and it is currently 2024, so the timeliness of the data is worth considering. In addition, what is the serial number of ethical approval?

Response to comments 9:

Firstly, we have added the serial number of ethical approval to the manuscript, which is 2021NO.07 for this study. 

Secondly, the reviewer has doubts about the timeliness of the data in this study. This study is a comprehensive series of studies on internet addiction, and so far, research on internet addiction is still ongoing. This study is an in-depth exploration and comprehensive analysis of preliminary data.

Revised content: 

See line:278 (revised manuscript with track changes)

See line:211 (manuscript)

Comments 10: The data is collected through the psychological system. Is there a valid link to this system?

Response to comments 10: 

We apologize for the wording error. This study used a paper questionnaire, which was distributed and collected offline. We have added the process of questionnaire collection for this study to the Procedures. Meanwhile, we have added some original paper questionnaires on the last page of this letter.

Revised content: 

See lines:322-326 (revised manuscript with track changes)

See lines:247-252 (manuscript)

Comments11: Please provide sample items for each scale.

Response to comments 11:

Thank you for your suggestion. We have attached sample items for Chinese version of each scale in Response to Reviewers, and have also included some paper questionnaires at the end of the letter.

Comments 12: Please specify the process of questionnaire collection and how to identify invalid questionnaires?

Response to comments 12:

Thank you for your valuable feedback and questions regarding the collection of this research questionnaire. This study contacted various professional psychological commissioners through the college psychological organization and collected paper questionnaires offline. Two psychology graduate students and one psychological commissioner from each major jointly distributed and collected questionnaires. The two psychology graduate students read out guidance before distributing the questionnaires, and after distribution, they were collected uniformly. 

Secondly, we have added the invalid questionnaire criteria the process of questionnaire collection for this study to the Procedures.

In this study, if the number of questions filled in the questionnaire is less than one-third, it will be considered invalid.

Revised content: 

See lines:322-326, 329-330. (revised manuscript with track changes)

See lines:247-252, 253-254. (manuscript)

Comments13: Authors only provide reliability, whether the validity meets the criteria, please add.

Response to comments 13:

Thank you for your suggestions on improving the manuscript. We have added the content of validity analysis for this study. This study analyzed the structural validity of each scale and added correlations based on the MI coefficient. Finally, it was shown that all pathways of each scale were statistically significant. Based on the statistical indicators proposed by Fornell et al., Doll WJ et al., and Hooper D et al., the final fit indices of the structural validity model in this study were acceptable. In this study, the final structural validity fit indices of each scale were as follow: maternal parenting style(χ2/df= 3.367, RMSEA=0.049, GFI=0.828, SRMR=0.090, TLI=0.843), paternal parenting style (χ2/df =3.449, RMSEA=0.050, GFI=0.820, SRMR=0.090, TLI=0.833), internet addiction (χ2/df=4.677, RMSEA =0.062, GFI=0.891, SRMR=0.046, TLI=0.879) and life satisfaction(χ2/df=4.439, RMSEA=0.032, GFI=0.992, SRMR=0.014, TLI=0.989).

The final structural model diagram is attached in Response to Reviewers.

The results have been added to the Measures of the manuscript; The reference indicators have been added to the Data Analysis.

Revised content:

CFA results:

See lines:296-298, 305-308, 316-319 (revised manuscript with track changes)

See lines:226-229, 233-236, 243-245 (manuscript)

Reference indicators:

See lines:339-343(revised manuscript with track changes)

See lines:263-266(manuscript)

Comments 14: Please provide valid references for the standard values of all indicators in the Results section, such as Common method deviation test (180 lines), etc.

Response to comments 14:

Thank you to the reviewer for providing valuable feedback. According to this suggestion, we have added references to the standards for statistical analysis of the manuscript. We have added reliability indicators, structural validity indicators, common method bias indicators, and references. All references for the standard values have been added to the Data Analysis. 

Revised content: 

See lines: 337-343, 346-348 (revised manuscript with track changes)

See lines: 262-266, 270-271 (manuscript)

Comments 15: Please add practical suggestions.

Response to comments 15:

Thank you for your suggestions on improving the manuscript. We tried our best to improve the manuscript and provided more specific suggestions. We have made modifications to the suggestions in the Discussion.

This study might be helpful for parents to rethink about their parenting styles, such as improving or adjusting their parenting behavior to reduce the risk of adolescent internet addiction. Parents should care about and support adolescents, develop positive interactions with them, praise them. Meanwhile, the influence of parenting styles on college students' internet addiction is also influenced by the factor of life satisfaction. This suggests that in the future, prevention and intervention for college students' internet addiction should help college students improve their own life satisfaction, for example, helping to improve emotional regulation, relax, and increase effective interpersonal communication. Moreover, when focusing on the work of both males and females, attention should be paid to the impact of different parenting dimensions on individuals’ internet addiction. In conclusion, this study suggests that in the future, more attention should be paid to the role of family factors in solving the internet addiction among college students. Starting from the cooperation between families and schools, the prevalence of internet addiction among college students may be reduced.

Revised content:

See lines: 574-583,587-592 (revised manuscript with track changes)

See lines: 477-485,492-496 (manuscript)

Comments 16: Please specify exactly how future research should proceed in the Limitations.

Response to comments 16:

Thank you for your suggestion about Limitations. We have carefully supplemented the content that could be improved for future research in Limitations. In Limitations, regarding the sampling issue, we think that in the future, we can expand the source of samples and collect samples from multiple universities and wider regions. Secondly, it is difficult to draw causal conclusions through cross-sectional studies in this study. In the future, a longitudinal tracking study can be used to improve the research.

Revised content: 

See lines: 599-600,602 (revised manuscript with track changes)

See lines: 500-501,503 (manuscript)

Comments 17: The findings, contributions and implications of this study need to be summarized in the concluding section.

Response to comments 17:

Thank you for your suggestions on improving the manuscript. We have carefully revised the Conclusion, integrating and expressing the objective of this study, as well as the results and recommendations obtained from the study. 

The Conclusion is as follows. This study mainly explored the potential mechanisms of internet addiction, the impact of family parenting styles on college students’ internet addiction, the mediating role of life satisfaction, and the role of gender in it. Parenting styles, particularly that of the father, have a major effect on students’ internet addiction, which is partially mediated by life satisfaction. For students of different genders, life satisfaction has different mediating effects, and the influence of parenting styles also present different forms. These findings indicate that paying attention to the role of the family, especially the parenting style of fathers, is crucial for preventing internet addiction in the future. This study may help parents rethink their parenting styles, such as improving or adjusting their parenting behavior based on gender, to reduce the risk of adolescent internet addiction.

Revised content: 

See lines: 609-611,615-619 (revised manuscript with track changes)

See lines: 510-512,516-520 (manuscript)

Finally, regarding the reviewer's suggestion on the reference doi, we have added the relevant literature's doi. 

See lines: 730,734,832,853 (revised manuscript with track changes)

See lines: 625,629,723,744 (manuscript)

#Reviewer 2:

Comments 1: No strong arguments for mediation are presented.

Response to comments 1:

Thank you for pointing out the research shortcomings. We think this is an excellent suggestion. In the Introduction, we did not express the relationship between the main variables in depth and the logic was not clear. Therefore, we readjusted and reorganized the relationship between the main variables. We have adjusted and revised the research review content in The mediating role of life satisfaction (lines 159-204; revised manuscript with track changes) of this study. In this part, we respectively elaborated the relationship between family parenting styles and life satisfaction, and the relationship between Internet addiction and life satisfaction. Then, based on the theoretical model of this study: the cognitive-behavioral model of pathological internet use, and previous studies on life satisfaction as a mediating factor, we proposed the mediation model hypothesis of this study. 

Revised content: 

See lines: 159-165, 172-180, 196-204 (revised manuscript with track changes)

See l

---

## [Decision Letter · Decision Letter 1]

2 Apr 2024

PONE-D-23-36156R1Effect of paternal-maternal parenting styles on college students' internet addiction of different genders: The mediating role of life satisfactionPLOS ONE

Dear Dr. Chen,

Thank you for submitting your manuscript to PLOS ONE. After careful consideration, we feel that it has merit but does not fully meet PLOS ONE’s publication criteria as it currently stands. Therefore, we invite you to submit a revised version of the manuscript that addresses the points raised during the review process.

Be sure to:in the results section, please add the coefficients before the p value for significance or remove the p. It is unusual to find only p values.Reply to reviewer feedback. Please submit your revised manuscript by May 17 2024 11:59PM.  If you will need more time than this to complete your revisions, please reply to this message or contact the journal office at plosone@plos.org. Please include the following items when submitting your revised manuscript:A rebuttal letter that responds to each point raised by the academic editor and reviewer(s). You should upload this letter as a separate file labeled 'Response to Reviewers'.A marked-up copy of your manuscript that highlights changes made to the original version. You should upload this as a separate file labeled 'Revised Manuscript with Track Changes'.An unmarked version of your revised paper without tracked changes. You should upload this as a separate file labeled 'Manuscript'.If applicable, we recommend that you deposit your laboratory protocols in protocols.io to enhance the reproducibility of your results. Protocols.io assigns your protocol its own identifier (DOI) so that it can be cited independently in the future. For instructions see: https://journals.plos.org/plosone/s/submission-guidelines#loc-laboratory-protocols. Additionally, PLOS ONE offers an option for publishing peer-reviewed Lab Protocol articles, which describe protocols hosted on protocols.io. Read more information on sharing protocols at https://plos.org/protocols?utm_medium=editorial-email&utm_source=authorletters&utm_campaign=protocols.

We look forward to receiving your revised manuscript.

Kind regards,

Fadwa Alhalaiqa

Academic Editor

PLOS ONE

Journal Requirements:

Reviewers' comments:

Reviewer's Responses to Questions

**Comments to the Author**

1. If the authors have adequately addressed your comments raised in a previous round of review and you feel that this manuscript is now acceptable for publication, you may indicate that here to bypass the “Comments to the Author” section, enter your conflict of interest statement in the “Confidential to Editor” section, and submit your "Accept" recommendation.

Reviewer #1: All comments have been addressed

Reviewer #2: (No Response)

2. Is the manuscript technically sound, and do the data support the conclusions?

Reviewer #1: Partly

Reviewer #2: Yes

3. Has the statistical analysis been performed appropriately and rigorously? 

Reviewer #1: Yes

Reviewer #2: Yes

4. Have the authors made all data underlying the findings in their manuscript fully available?

Reviewer #1: No

Reviewer #2: No

5. Is the manuscript presented in an intelligible fashion and written in standard English?

Reviewer #1: Yes

Reviewer #2: Yes

6. Review Comments to the Author

**Reviewer #1:** Dear author,

Many thanks for your efforts in revising the manuscript. The revised manuscript is greatly improved in all respects. My opinion is that the manuscript is acceptable. But there are still two problems that the authors need to be aware of. First, “This study is a comprehensive series of studies on internet addiction, and so far, research on internet addiction is still ongoing. ” This reason does not explain the timeliness of the data, and cross-sectional studies should also consider the timeliness of the data. Second, the authors should have taken a more scientific approach to identifying invalid questionnaires. Please study further.

**Reviewer #2:** The authors addressed all my concern and the manuscript is significantly improved. I have only some small remarks.

I recommend to remove the heading Theory in the Introduction.

Avoid causal terms (e.g., the word influence in hypotheses).

The hypotheses could be presented at a specific level, by mentioning the specific mediation expected.

In the results section, please add the coefficients before the p value for significance or remove the p. It is unusual to find only p values.

7. PLOS authors have the option to publish the peer review history of their article (what does this mean?). If published, this will include your full peer review and any attached files.

Reviewer #1: No

Reviewer #2: **Yes: **Cornelia Mairean

---

## [Author Response · Author response to Decision Letter 1]

9 Apr 2024

Dear Editor and Reviewers,

We appreciate the thoughtful comments and constructive feedback provided by the Editor and Reviewers. Those comments and feedback are all valuable and very helpful for revising and improving our manuscript. According to the comments and feedback, we have made extensive modifications to the manuscript. The reviewers’ comments are laid out below in bold font and specific concerns have been numbered. And changes/additions to the manuscript are identified using the “Track Changes” option in Microsoft Word.

# Editor:

Comments 1. in the results section, please add the coefficients before the p value for significance or remove the p. It is unusual to find only p values.

Response to Comments 1:

Firstly, thank you for taking the time to review our manuscript and your reminder. We have carefully read and responded to every suggestion and evaluation from reviewers. 

For the first suggestion, as the reference values are already listed in the table. Therefore, we have carefully considered and removed the p-value from the Result of the manuscript.

Revised content: 

see lines 285-290, 303-311 (revised manuscript with track changes)

see lines 278-283, 296-303 (manuscript)

Comments 2. Reply to reviewer feedback.

Response to Comments 2:

In response to the two suggestions from #Reviewer 1

Comments 1. First, “This study is a comprehensive series of studies on internet addiction, and so far, research on internet addiction is still ongoing. ” This reason does not explain the timeliness of the data, and cross-sectional studies should also consider the timeliness of the data. 

Response to comments 1:

We gladly accept and will try our best to avoid the issues of data timeliness. In future research, we will pay attention to the timeliness of data in cross-sectional studies to avoid such issues from happening again. With new data, we will organize and analyze the data in a timely manner.

Comments 2. Second, the authors should have taken a more scientific approach to identifying invalid questionnaires. Please study further.

Response to comments 2:

Thank you to the reviewer for letting us discover the shortcomings in data processing. In the future, we will refer to previous research and most of the methods used by scholars to handle and identify invalid questionnaires more cautiously and scientifically.

In response to the suggestions from #Reviewer 2 

We have carefully made revisions to the manuscript raised by Reviewer 2, regarding the details of the manuscript.

Comments 1: I recommend to remove the heading Theory in the Introduction.

Response to comments 1:

We are very grateful for the details the reviewer has raised. We have removed the heading Theory in the Introduction and combined it with the rest to make the logic of the manuscript more coherent. 

Revised content: 

See lines: 72-80 (revised manuscript with track changes)

See lines: 65-73 (manuscript)

Comments 2: Avoid causal terms (e.g., the word influence in hypotheses).

Response to comments 2:

We have carefully reviewed the manuscript and revised the causal terms related to the viewpoints proposed in this study and replaced it with words such as association, relationship, and correlation.

Revised content:

See lines: 23, 26-27, 29, 112, 143, 190, 202, 324, 347, 351, 396, 401, 417, 426, 435, 437, 463, 468, 486(revised manuscript with track changes)

See lines: 23, 26, 29, 106, 137, 184, 195, 316, 339, 343, 388, 393, 408, 416, 425, 427, 453, 458, 476(manuscript)

Comments 3: The hypotheses could be presented at a specific level, by mentioning the specific mediation expected. 

Response to comments 3:

We have made specific statements about the assumptions proposed in the manuscript. We have refined the relationship between different types of parenting styles and internet addiction, assuming that the emotional warmth of parenting styles may be positively correlated with internet addiction, while parenting styles are all as overprotection, involvement, punishment, and preference are positively correlated with internet addiction.

Revised content: 

See lines: 199-203 (revised manuscript with track changes)

See lines: 192-196 (manuscript)

Comments 4: In the results section, please add the coefficients before the p value for significance or remove the p. It is unusual to find only p values.

Response to comments 4:

After careful consideration, regarding the p-value in the manuscript, as the reference values have been presented in the table. Therefore, we have carefully considered and removed the p-value from the Result of the manuscript.

Revised content: 

See lines: 285-290, 303-311 (revised manuscript with track changes)

See lines: 278-283, 296-303 (manuscript)

#Reviewer 1:

Dear author,

Many thanks for your efforts in revising the manuscript. The revised manuscript is greatly improved in all respects. My opinion is that the manuscript is acceptable. But there are still two problems that the authors need to be aware of. 

Response to #Reviewer 1:

Thank you for taking the time to provide valuable suggestions for improving the manuscript. Your suggestions have greatly helped to improve the research level and ideas of the researchers in this study. Moreover, we are glad to hear your affirmation of the manuscript. We will keep in mind your two remaining questions about the manuscript and consciously avoid such problems from happening again in future research. 

Comments 1. First, “This study is a comprehensive series of studies on internet addiction, and so far, research on internet addiction is still ongoing. ” This reason does not explain the timeliness of the data, and cross-sectional studies should also consider the timeliness of the data. 

Response to Comments 1: 

Thank you for your suggestions and feedback on data timeliness. In future research, we will pay attention to the timeliness of data in cross-sectional studies to avoid such issues from happening again. With new data, we will organize and analyze the data in a timely manner.

Comments 2. Second, the authors should have taken a more scientific approach to identifying invalid questionnaires. Please study further.

Response to Comments 2:

Thank you for letting us discover the shortcomings in data processing. In the future, we will refer to previous research and most of the methods used by scholars to handle and identify invalid questionnaires more cautiously and scientifically.

#Reviewer 2:

The authors addressed all my concern and the manuscript is significantly improved. I have only some small remarks.

Comments 1: I recommend to remove the heading Theory in the Introduction.

Response to comments 1:

Thank you for your affirmation of our revised manuscript. We are very grateful for the details you have raised, and these suggestions can help improve the manuscript even better. For Comments 1, we have removed the heading Theory in the Introduction and combined it with the rest to make the logic of the manuscript more coherent.

Revised content: 

See lines: 72-80 (revised manuscript with track changes)

See lines: 65-73 (manuscript)

Comments 2: Avoid causal terms (e.g., the word influence in hypotheses).

Response to comments 2:

As suggested by the reviewer, we have carefully reviewed the manuscript and revised the causal terms related to the viewpoints proposed in this study and replaced it with words such as association, relationship, and correlation.

Revised content:

See lines: 23, 26-27, 29, 112, 143, 190, 202, 324, 347, 351, 396, 401, 417, 426, 435, 437, 463, 468, 486(revised manuscript with track changes)

See lines: 23, 26, 29, 106,137, 184, 195, 316, 339, 343, 388, 393, 408, 416, 425, 427, 453, 458, 476(manuscript)

Comments 3: The hypotheses could be presented at a specific level, by mentioning the specific mediation expected. 

Response to comments 3:

We agree with the reviewer’s suggestion and have incorporate the recommended changes into manuscript. We have made specific statements about the assumptions proposed in the manuscript. We have refined the relationship between different types of parenting styles and internet addiction, assuming that the emotional warmth of parenting styles may be positively correlated with internet addiction, while parenting styles are all as overprotection, involvement, punishment, and preference are positively correlated with internet addiction.

Revised content: 

See lines: 199-203 (revised manuscript with track changes)

See lines: 192-196 (manuscript)

Comments 4: In the results section, please add the coefficients before the p value for significance or remove the p. It is unusual to find only p values.

Response to comments 4:

Thank you for your valuable feedback on the details of the manuscript. After careful consideration, regarding the p-value in the manuscript, as the reference values have been presented in the table. Therefore, we have carefully considered and removed the p-value from the Result of the manuscript.

Revised content: 

See lines: 285-290, 303-311 (revised manuscript with track changes)

See lines: 278-283, 296-303 (manuscript)

Firstly, all researchers in this study are very grateful for the rigorous review of the article by the Editor and the Reviewers. We sincerely appreciate the time and effort invested by the Reviewers and Editor in evaluating our manuscript. We look forward to any additional feedback or suggestions.

---

## [Decision Letter · Decision Letter 2]

29 Apr 2024

Effect of paternal-maternal parenting styles on college students' internet addiction of different genders: The mediating role of life satisfaction

PONE-D-23-36156R2

Dear Dr. Zi Chen,

We’re pleased to inform you that your manuscript has been judged scientifically suitable for publication and will be formally accepted for publication once it meets all outstanding technical requirements.

Kind regards,

Fadwa Alhalaiqa

Academic Editor

PLOS ONE

Additional Editor Comments (optional):

Reviewers' comments:

Reviewer's Responses to Questions

**Comments to the Author**

1. If the authors have adequately addressed your comments raised in a previous round of review and you feel that this manuscript is now acceptable for publication, you may indicate that here to bypass the “Comments to the Author” section, enter your conflict of interest statement in the “Confidential to Editor” section, and submit your "Accept" recommendation.

Reviewer #2: All comments have been addressed

2. Is the manuscript technically sound, and do the data support the conclusions?

Reviewer #2: Yes

3. Has the statistical analysis been performed appropriately and rigorously? 

Reviewer #2: Yes

4. Have the authors made all data underlying the findings in their manuscript fully available?

Reviewer #2: Yes

5. Is the manuscript presented in an intelligible fashion and written in standard English?

Reviewer #2: Yes

6. Review Comments to the Author

Reviewer #2: (No Response)

7. PLOS authors have the option to publish the peer review history of their article (what does this mean?). If published, this will include your full peer review and any attached files.

Reviewer #2: **Yes: **Cornelia Mairean

---

## [Editor Report · Acceptance letter]

2 May 2024

PONE-D-23-36156R2 

PLOS ONE

Dear Dr. Chen, 

I'm pleased to inform you that your manuscript has been deemed suitable for publication in PLOS ONE. Congratulations! Your manuscript is now being handed over to our production team.

Kind regards, 

on behalf of

Pro Fadwa Alhalaiqa 

Academic Editor

PLOS ONE